# Dynamic Evolution of *NLR* Genes in Dalbergioids

**DOI:** 10.3390/genes14020377

**Published:** 2023-01-31

**Authors:** Shamiza Rani, Ramlah Zahra, Abu Bakar, Muhammad Rizwan, Abu-Bakar Sultan, Muhammad Zain, Amna Mehmood, Muhammad Danial, Sidra Shakoor, Fozia Saleem, Ali Serfraz, Hafiz Mamoon Rehman, Rao Sohail Ahmad Khan, Saad Serfraz, Saad AlKahtani

**Affiliations:** 1Evolutionary Biology Lab, Centre of Agricultural Biochemistry and Biotechnology (CABB), University of Agriculture, Faisalabad 38000, Pakistan; 2Metabolomics Innovative Insitute, University of Alberta, Edmonton, AB T6G 2R3, Canada; 3Department of Plant Pathology, University of Arid Agriculture, Rawalpindi 46000, Pakistan; 4Cotton Genomics Lab, Centre of Agricultural Biochemistry and Biotechnology (CABB), University of Agriculture, Faisalabad 38000, Pakistan; 5Department of Zoology, College of Science, King Saud University, P.O. Box 2455, Riyadh 11451, Saudi Arabia

**Keywords:** *NLR* genes, wild *Arachis* species, allotetraploids, allopolyploid, NLR expansion, contraction, R genes gain and loss, novel resistance resources, peanut

## Abstract

*Dalbergioid* is a large group within the family *Fabaceae* that consists of diverse plant species distributed in distinct biogeographic realms. Here, we have performed a comprehensive study to understand the evolution of the nucleotide-binding leucine-rich repeats (NLRs) gene family in *Dalbergioids*. The evolution of gene families in this group is affected by a common whole genome duplication that occurred approximately 58 million years ago, followed by diploidization that often leads to contraction. Our study suggests that since diploidization, the NLRome of all groups of *Dalbergioids* is expanding in a clade-specific manner with fewer exceptions. Phylogenetic analysis and classification of NLRs revealed that they belong to seven subgroups. Specific subgroups have expanded in a species-specific manner, leading to divergent evolution. Among the *Dalbergia* clade, the expansion of NLRome in six species of the genus *Dalbergia* was observed, with the exception of *Dalbergia odorifera,* where a recent contraction of NLRome occurred. Similarly, members of the *Pterocarpus* clade genus *Arachis* revealed a large-scale expansion in the diploid species. In addition, the asymmetric expansion of NLRome was observed in wild and domesticated tetraploids after recent duplications in the genus *Arachis*. Our analysis strongly suggests that whole genome duplication followed by tandem duplication after divergence from a common ancestor of *Dalbergioids* is the major cause of NLRome expansion. To the best of our knowledge, this is the first ever study to provide insight toward the evolution of *NLR* genes in this important tribe. In addition, accurate identification and characterization of *NLR* genes is a substantial contribution to the repertoire of resistances among members of the *Dalbergioids* species.

## 1. Introduction

*Dalbergioids* are one of the largest clades in the subfamily *Papilionoideae* of the family *Fabaceae*. Members of this group have pantropical distribution, and a majority of them are restricted to neotropical regions [1]. It consists of 44 genera and 1383 species [1,2]. According to taxonomic distribution, *Dalbergioids* are divided into three major clades (1) *Adesmia* clade, which comprises herbaceous plants (2) *Dalbergia,* and (3) *Pterocarpus,* which consists of trees, shrubs, and woody lianas [1,2]. Polyploidy and dysploidy are quite common phenomena of *Dalbergioids*; three important genera, *Arachis*, *Stylosanthes,* and *Aeschynomene* (Ae)*,* show the mechanism of polyploidy [2]. Frequent polyploidy and dysploidy are the major reasons for the lack of precise base chromosome numbers, and a recent analysis identified a base number of 10 chromosomes for a member of *Dalbergioids* [2].

*Dalbergioids* consist of a great deal of ecological and economically important members, especially *Arachis* and *Dalbergia*. *Arachis* is a major legume crop that is grown on 25 million ha with an annual production of ~46 million tons [3,4]. Its center of origin is in South America, where *Arachis hypogaea* were domesticated ~6000 years ago and then widely distributed globally during post-Columbian times [5]. Similarly, *Dalbergia* species are also economically important crops as they are the valuable source of heartwood timber, known as rosewood, and are incorporated in a wide range of products. They are distributed pantropically across the Americas, Africa, and Asia [6]. According to the International Union for Conservation of Nature (IUCN), many species of the genus *Dalbergia* are classified as vulnerable or endangered. In order to regulate the international trade of *Dalbergia* timber and prevent illegal harvesting, the whole genus of *Dalbergia* was listed in the Convention on International Trade in Endangered Species (CITES). In addition, the genus *Dalbergia* is also threatened due to the wide range of biotic and abiotic stress factors. One of the important species, *Dalbergia sissoo,* which is the main source of timber in South East Asia, is threatened by several root pathogens and extreme disease of dieback. It causes thinning, the drying up of leaves and branches, and the drying up of crown regions, and has been a major cause of large-scale tree mortality in South East Asia [7,8]. To this date, the disease etiology of dieback is not known. It is important to understand the evolution of disease-resistance genes among species of *Dalbergioids*. Gaining an understanding of their molecular mechanisms and their accurate detection and characterization is vital for combating biotic stress aspects.

Nucleotide-binding site leucine-rich repeat receptors (NLRs) recognize the pathogen’s effector via direct or indirect interaction, which activates a number of defensive mechanisms, one of which is a hypersensitive response, also known as localized programmed cell death [9]. NLRs mainly consist of NB-ARC and C-terminal leucine-rich repeats (NLRs). The NB-ARC domain is the most conserved region to determine the evolutionary relationship between plant NLRs [10]. There are four major classes of plant NLRs with distinct N-terminal domain fusions: (1) the TIR-NLR subclade containing an N-terminal Toll/interleukin-1 receptor (TIR) domain, (2) the CC-NLR subclade containing an N-terminal type Rx-type coil (CC) domain, (3) the CCR-NLR subclade containing the RTP8-type CC domain, and the recently proposed (4) G10 subclade that contains the distinct type of CC and forms a monophyletic group. Genome-wide identification and annotation of *NLR* genes from plants are challenging, owing to their complex sequence diversity and evolutionary history. However, the recently released tool, NLRtracker, identifies and characterizes *NLR* genes in a high-throughput manner using canonical features of functionally characterized plant resistance genes [11].

Here, we attempt to understand the evolution of *NLR* genes in members of *Dalbergioids* using the assembled genome of *Nissolia schottii*, *Aeschynomene evenia*, *Dalbergia odorifera*, and *Dalbergia sissoo,* and six species of the genus *Arachis*. We also screened five additional species of *Dalbergia* using their reference transcriptomes [12]. Here, we have addressed an important question regarding the evolution of *NLR* genes in this diverse group. Whether the members of the *Dalbergieae* tribe have followed the same global trend, or have they evolved in a clade-specific manner? Does the genome size have a correlation with the NLRome among *Dalbergioids*? What are the potential wild species that can be used as potential resistance resources? Additionally, how does the phenomenon of polyploidy affect the evolution of *NLR* genes among the members of *Dalbergioids*?

## 2. Materials and Methods

### 2.1. Mining of NLR Genes in Arachis Species

The assembled transcriptomes of six *Dalbergioids* species were downloaded from the NCBI database (Appendix A). Genome, complete coding sequence (CDS), and reference proteome files for *Arachis duranensis*, *A. ipaensis*, *A. hypogaea*, *A. cardenasii*, *A. stenosperma*, *A. monticola*, *D. odorifera*, *N. schottii,* and *Ae. evenia* were acquired from NCBI database (https://www.ncbi.nlm.nih.gov/genome/, accessed on 1 January 2023). The genome of *Dalbergia sissoo* was downloaded from the 10K genome portal (https://db.cngb.org/10kp/, accessed on 10 November 2022) and was later annotated using Augustus (v-3.4.0) [13], with default settings except for the option of complete gene models (--genemodel=complete). The resulting GFF file was parsed into amino acids and coding sequences using two Perl scripts (getAnnotFasta.pl and gffread) [13]. Reference transcriptomes of five additional *Dalbergia* species were downloaded from transcriptome shotgun assembly (TSA: https://www.ncbi.nlm.nih.gov/Traces/wgs, accessed on 10 November 2022). Reference proteomes and transcriptomes were subjected to the NLRtracker pipeline, which extracts and annotates NLRs from proteins and transcript files. NLRtracker pipeline uses Interproscan [14] and predefined NLR motifs [15] to extract NLRs and provide domain architecture analyses based on the canonical features found in reference plant *NLR* genes. NLRtracker annotation of CCR-NLR remained undetermined; for this reason, manual curation was performed for each *NLR* gene using clustering and phylogenetic analysis.

### 2.2. Clusterization and Phylogenetic Analysis

A library of NB-ARC domain was constructed from reference *NLR* genes of the PRG database [10] and clustered using UCLUST [16], with an identity threshold of 70%. The resulting reference genes from each cluster were classified into subgroups that were already defined by Eunyoung Seo et al. [17], and were considered as seed probes for phylogenetic and clustering analysis. For comprehensive phylogenetic analysis, extracted NB-ARC domains (output of NLRtracker) from each species were aligned with seed probes of NB-ARC using MUSCLE (version 1.26, Hull, 2009). Subsequent maximum likelihood analysis was performed using IQtree v 2.0 [18], choosing the best-fit model of evolution (m JTT+F+R10) and 1000 bootstrap replicates.

### 2.3. Chromosomal Localization and Construction of a Syntenic R-Gene Maps

Coordinates for identified *NLR* genes were extracted and subjected to density distribution analyses. Unplaced scaffolds were excluded, and chromosomal contigs were considered for binning. The number of NLR homologs in 5 Kb bins of each *Dalbergioid* genome was obtained using “make-windows” and “intersect” commands of the BEDTools program [19]. Each bin was then manually labeled with a serial number. Using bin number and NLR density value in each bin, a linearized version of the genome was visualized using the Rideogram package [20]. To find the syntenic relationship between NLRs in *Dalbergioids* species, respective BED files from each species (bin size = 5 kb) were used for the initialization of genomic tracks. BLAST was performed for identification of inter-specie genomic similarities, then chromosome and genomic position were retrieved from the GTF file and subsequently sorted according to BLAST output. Genomic linkage was provided on collinearity bases between the genes. The R package, “Circlize” [21], was used for the visualization of synteny plots.

### 2.4. Evolutionary Analysis of NLRs

Clustalw was used to align each group of paralogs’ deduced protein sequences across their respective subgroups [22]. Additionally, the obtained alignment was used as a guide in order to align corresponding nucleotide sequences via the usage of the pal2nal software (version 1.0), which is based on the language Perl [23]. After removing gaps and N-coding codons, ks were estimated using ka/ks calculator under the MA method [24]. We performed the Fisher test on each paralog selection value, and significant duplication events were kept, and the rest of them were removed (*p* value > 0.01). Ks values greater than two (>2) were eliminated from further consideration, since there is a possibility that they suggest substitution saturation. Orthovenn2 [25] was utilized to study orthologs cluster *NLR* genes. Identified putative *NLR* genes from each species were queried in a locally installed Orthovenn2 program (version 2.1) using an E-value of 1 × 10^−2^ with default settings. All *NLR* genes identified were subjected to Orthofinder analysis [26]. Output containing orthogroup families was labeled manually, and the species tree was modified into an ultrametric tree using the R package ape [27]. Both files were utilized as input for the CAFE5 [28], and the resulting files were manually parsed to evaluate gene gain and loss at each node of the species’ phylogenetic tree. Furthermore, the ortholog sequences between each species were also acquired to study the selection rate using Orthofinder [26].

### 2.5. RNA-Seq Based Expression Analysis

Basal expression level of NLRs identified from this study was evaluated using the available datasets of *D. odorifera* (Appendix A). The first dataset provides comprehensive collection of replicates of the root, leaf, stem, flower, and seeds (PRJNA593817). In the current study, we aligned the raw read sequences using the reference genome of *D. odorifera* with HISAT [29]. Alignments were passed to StringTie [29] for transcript assembly. Finally, the assembled transcripts and abundances were processed using Ballgown [29] for grouping of experimental conditions and the determination of the differences expressed between the conditions. Similarly, we also evaluated the expression of *NLR* genes in 10 species of genus *Aeschynomene* at the time of root nodulation using a BioProject (PRJNA459484). All expression values (FPKM) of all genes expressed in one species were summated to calculate the cumulative expression value as a marker to study the quantitative expression. Secondly, average number of *NLR* genes expressed in each species was also calculated. For genus *Arachis*, we evaluated the expression of *NLR* genes in progenitors and their allotetraploid species using a dataset from BioProject PRNA380954.

## 3. Results

### 3.1. Gene Mining of NLR Genes in Dalbergioids Species

In this study, we employed the NLRtracker pipeline for the identification and annotation of NLRome in members of *Dalbergioids*. Expanded NLRome was observed in the case of genus *Arachis* (*Pterocarpus* clade), where, in total, 310, 290, 392, 644, 660, 490, 547, and 817 *NLR* genes were identified from *A. duranensis* and *A. ipaensis*, *A. monticola* (AA), *A. monticola* (BB), *A. hypogea* (AA), *A. hypogea* (BB), *A. stenosperma*, and *A. cardenasii*, respectively (Figure 1). Furthermore, 180 *NLR* genes were identified from the *Adesmia* clade from the member *N. schottii*. A highly variable number of *NLR* genes was observed from the *Dalbergia* clade, with 113, 83, 251, 180, 181, 221, 228, 154, and 206 from *A. evenia*, *D. odorifera*, *D. sissoo*, *D. frutescens*, *D. cochichinensis*, *D. oliveri,* and *D. melanoxylon*, respectively. The redundancy or duplication of the NLR homolog from the assembled transcriptome of the genus *Dalbergia* was removed by clustering *NLR* genes at 70% sequence identity; then, a representative gene from each cluster was considered as a single *NLR* gene. Interestingly, *D. odorifera* still has shown the least number of *NLR* genes compared to the other *Dalbergioids* species. This dramatic contraction of NLRome in *D. odorifera* cannot be explained comprehensively, due to the lack of reference genomes from other *Dalbergia* species. All four classes of the *NLR* genes were present in all members of *Dalbergioids*. Overall, class TIR-NLR exhibits the highest contribution among the other classes, with an average of 46% TIR-NLR. Similarly, other genes represented 40% CC-NLR and 23% CCG10-NLR. A variable number of helper *NLR* genes were observed, ranging from 1 to 5%. The average length of NLR ranged between 500 and 600 amino acids, and the average NB-ARC length ranged from 200 to 300 amino acids among *Dalbergioids*.

### 3.2. Genomic Localization of NLRs among Dalbergioids

A total of 5353 *NLR* genes were mapped on the linearized chromosomal map of the Dalbergioids species (Figure 2). Two genomes, *N. schottii* and *D. sissoo,* were not included in this analysis, due to the lack of chromosomal-anchored genome assembly. Gene density maps suggest a lack of an apparent positive correlation with the individual genome size in the clade *Dalbergia*. Where *A. evenia* with the smallest genome size of 376 Mb [30] showed an expanded NLRome, as compared *D. odorifera* (653 Mb). Similarly, *D. sissoo* had a genome size of 640 Mb and also showed an expanded NLRome, as compared to *D. odorifera* (Figure 2). In the case of the *Pterocarpus* clade, the *Arachis* species with an average genome size of 1222.2 Mb revealed an overall scattered gene density. Interestingly, the lack of apparent correlation between the genome size and the number of *NLR* genes was also vibrant. *A. cardenasii,* with a genome size of 1131 Mb, showed the most expanded NLRome as compared to *A. hypogaea* (B-subgenome), with 1444 Mb, which showed a contracted NLRome. In addition, significant synteny was observed between clusters of *NLR* genes in the Arachis species. In short, the expansion and contraction of NLRome in *Dalbergioids* are not directly linked with plasticity in the genome size.

### 3.3. Phylogenetic Distribution

The conserved NBARC domain was extracted from each *Dalbergioids* species and clustered at a 75% sequence identity using CD-HIT [31]. The representative members from each cluster were utilized for the reconstruction of the phylogenetic relationships among *N. schottii*, *A. evenia*, *D. odorifera*, *D. sissoo*, *D. frutescens*, *D. cochichinensis*, *D. oliveri, D. melanoxylon*, *A. duranensis*, and *A. ipaensis* (Figure 3). The TNL clade branched out as expected and remained polyphyletic with four radiations. On the other hand, the CNL clade was divided into three monophyletic major sub-clades CC-NLR, CCR-NLR, and CCG10-NLR. CC-NLR was further divided into four major sub-groups CNL-Un, CNL-G11, CNL-G7, and G4. Significant diversity and expansion were observed in CCG10-NLR, as compared to other members of the family *Fabaceae,* especially in the case of *N. schottii*. Overall, two major sub-clades were observed for the CCG10-NLR subgroup. Similarly, G4 and G7 showed expansion and diversity, which is typical of *Fabales*. Multiple polyphyletic radiations were observed for G7, whereas G4 remained strictly monophyletic among all of the *Dalbergioids* member species. Interestingly, CNL groups G1, G2, G3, G4, G6, and G8, previously identified from the *Solanaceae* family, were absent in *Dalbergioids*. That is consistent with studies on the genus *Cicer*, which strongly suggest that *Fabaceae* members lack G1-G8 groups [32]. Interestingly, the highest number of TIR and CC-*NLR* genes were observed among members of the clade *Petrocarpus* (*Arachis*) and *Adesmia* (*Nissolia*). These unbalanced gene duplication occurrences suggest a possible role of terminal duplication after divergence from the common ancestor of *Dalbergioids*.

### 3.4. The Birth and Death of NLR Genes in Dalbergioids

NLRome shows a rapid contraction and expansion, which is directly linked with the extent of the pathogen interaction [11]. As indicated earlier, remarkable differences among a number of *NLR* genes were observed in *Dalbergioids*. In this study, the conserved and species-specific *NLR* genes were studied in eight members of *Dalbergioids*. Interestingly, reduced conservation of *NLR* genes was found in all members, and only 24 gene clusters were found to be conserved (Figure 4). However, members belonging to similar clades exhibited an increased ratio of conservation of *NLR* genes among themselves. It suggests that *NLR* genes were expanded in a clade-specific manner.

We constructed a phylogenetic tree using CAFÉ [28] and mapped the gene gain and loss events. High birth and death rates after the divergence from a common ancestor *of Dalbergioids* suggests dynamic expansion and contraction (Figure 4). For example, *A. evenia* exhibited dynamic contraction of NLRome with a higher rate of the gene death ratio. On the other hand, *N. schottii* from the *Adesmia* clade revealed the highest rate of terminal gene duplication (108) (Figure 4). Similarly, members of the genus *Dalbergia* have shown a net gene gain during the process of speciation and a considerable rate of terminal duplication with the exception of *D. odorifera,* where a dramatic contraction of NLRome was observed in nearly 38 gene families that were lost. The frequent births and deaths of *NLR* genes in *Dalbergioids* suggest their highly distinct evolutionary pattern.

In the case of the clade *Petrocarpus*, polyploidy played a major part in the evolution of the *NLR* gene family in the *Arachis* species. Consistent with previous studies [33], an asymmetric evolution of *NLR* genes was observed in tetraploid species (*A. monticola* and *A. hypogaea*). Gene gain and loss analysis confirms the expansion of NLRome in the A-subgenome of *A. hypogaea,* whereas it confirms contractions in wild tetraploid species *A. monticola* (A-subgenome). Contrastingly, the B-subgenome revealed an expansion in *A. monticola* and a contraction in *A. hypogaea* (Appendix A). Similarly, wild species *A. cardenasii* and *A. stenosperma* also showed an expansion of NLRome with gains in gene families. The tandem duplication after speciation also played an important role in the expansion of NLRome, and large-scale terminal duplications can be observed in *A. cardenasii*, *A. stenosperma,* and *A. monticola* (B-subgenome).

### 3.5. Duplication History among Dalbergioids NLRs

We further studied the expansion history of *Dalbergioids*. The gene duplication time was estimated by computing ks between genes within the same subgroup. The divergence time of all three clades of *Dalbergioids* is estimated to be ~38 mya (million years ago) based on the ks value of 0.6 (reference of *Arachis* divergence time and substitution rate). The distribution of Ks values between NLR paralogs in *N. schottii* peaked between 0.32 and 0.64 (19 to 36 mya), indicating that the duplication events would have occurred after the speciation from a common ancestor. In addition, each subgroup showed a distinctive pattern of duplication. Subgroups CCG10-NLR, G4-NLR, G7-NLR, and TIR-NLR showed the duplication increase after speciation, whereas reduced duplications were observed in G11, CNL-Un, and CCR-NLR (Figure 5).

NLR Ks values from each species were compared. Two species, *N. schottii,* and *A. duranensis,* exhibited similar patterns of gene duplication and peaked between 0.16 and 0.64, whereas the *A. evenia* and *A. ipaensis* showed a peak between 0.08 and 0.16. It should be noted that the progenitor species of *Arachis hypogea* (*A. duranensis* and *A. ipaensis*) showed a difference in duplication history, since *A. duranensis* represents A-genome and *A. ipaensis* represent B-subgenome, which might be the reason for the distinct NLR duplication history. In addition, *N. schottii* CCG10-NLR exhibited large-scale duplication after speciation. Surprisingly, both species of the genus *Dalbergia* showed the lowest ratio of gene duplication (Figure 5). This unequal gene duplication pattern has led to the evolution of different gene repertoires.

### 3.6. Expression in NLR Genes in Dalbergioids

Furthermore, we also evaluated whether the identified *NLR* genes are transcriptionally active, and if they were, up to what extent were they are active. Here, we studied the expression of *NLR* genes across different species of clade *dalbergia* and petrocarpus. Among Dalbergia, the majority of *NLR* genes were across five different species of this genus: *D. sissoo*, *D. frutescens*, *D. cochichinensis*, *D. oliveri,* and *D. melanoxylon* using a dataset from PRJNA593817. We also compared the expression of *NLR* genes in different tissues of *D. odorifera*. In total, 20 *NLR* genes were expressed in all types of tissue, including in the root, leaf, flower, stem, and seeds (Appendix A).

*NLR* genes play a key role in the recognition of root nodulation causing bacteria. Here, we studied the expression of *NLR* genes at the time of root nodulation in 10 distinct species of *Aeschynomene genus.* Interestingly, on average, 50 *NLR* genes were expressed with varying degrees of expression in distinct species of *Aeschynomene*. Furthermore, *A. ciliate*, *A. rubis*, and *A. denticulata* revealed higher qualitative and quantitative expressions of *NLR* genes (Figure 6A–C). We also evaluated the expression of *NLR* genes in progenitor species *A. duranensis*, *A. ipaensis,* and their allotetraploid *A. monticola* and *A. hypogaea*. On average, all species showed an expression of 39 *NLR* genes, with a higher cumulative expression in *A. monticola*. In short, the *NLR* genes identified in this study from *Dalbergioids* are transcriptionally active and may play an important role in the host’s stress response (Appendix A).

## 4. Discussion

Whole genome duplication is one of the reasons for the unique plasticity of plant genomes. It allows duplication and diversification of protein-coding genes and especially quantitative genes, such as transcriptions factors and microRNA genes [34,35]. About 60 mya, the *Fabaceae* family underwent a single, whole genome duplication, during which the rapid expansion of *NLR* genes occurred, followed by diploidization 20 mya, which caused a large contraction of *NLR* genes [36]. All major branches of the family *Fabaceae* suffered from a contraction 40–60 mya, including the common ancestor of *Dalbergioids,* which is consistent with the outcome of the current study that suggests the presence of only 24 core *NLR* gene clusters in the common ancestor. However, the expansion of NLRome occurred after divergence. Here, we identified the expansion of NLRome in *N. schottii*, the genus *Dalbergia* and *Arachis*. The net gain of gene clusters and pronounced species-specific tandem duplications were the major mechanism for the expansion of NLRome among *Dalbergioids*. However, the species-specific dramatic contraction was also observed in the case of *Ae. evenia,* where 46 gene clusters were lost during the speciation.

We analyzed seven species of the genus *Dalbergia* that contain extant species endemic to Asia. Our analysis suggests that the NLRome of the basal species *D. miscolobium* revealed continuous contractions until the split between 11 and 13 mya. After further divergence from the basal *Dalbergia* species, there was a gradual expansion through the birth of gene clusters in *D. oliveri*, *D. cochinchinensis*, *D. sissoo*, *D. melanoxylon,* and *D. frutescent*. Here, tandem duplication played an important role in increasing the number of *NLR* genes. However, a dramatic contraction was observed in *D. odorifera*, where the rapid loss of 38 *NLR* gene clusters occurred during its speciation. The recently published genome of *D. odorifera* also suggests the overall global contraction of gene families in *D. odorifera,* and in total, 214 gene families were lost during its speciation [36]. *D. odorifera* is an endemic species from the island of Hainan, China, and it has substantial medical importance. This remarkable contraction of NLRome may be due to its speciation in an insulated environment of the island of Hainan [37]. It suggests that *D. odorifera* is an atypical member of the genus *Dalbergia*. Our study suggests that the *Dalbergia* species shows a consistent expansion of protein-coding genes from the basal species, contrary to *D. odorifera*. The availability draft genome and additional reference transcriptome is imperative for further understanding of the evolutionary aspects of this genus.

Polyploidy is another type of whole genome duplication that significantly affects the gene dosage and leads to divergent evolution [4,38]. The distinct evolution of gene families during post-duplication scenarios may cause their contraction or expansion. In the case of the genus *Arachis* progenitor and other diploid species, they showed a gradual accumulation of *NLR* genes, and *A. cardenasii* has the largest expanded NLRome among all the other species. Here, we also observed that allotetraploids were kept and survived due to their increased gene dosage, which affected the survival of species by accumulating useful traits. Wild allotetraploid *A. monticola* was domesticated and led to the evolution of *A. hypogaea* [3,4]. Both domesticated and wild tetraploids showed an asymmetric expansion of NLRome. This distinct evolution of A and B subgenomes might be due to the presence of artificial selection in domesticated species and pronounced natural selection in wild tetraploids. Both tetraploids suggest the expansion of the *NLR* gene family is consistent with an asymmetric expansion of other gene families that are involved in starch and sucrose metabolism, linoleic acid metabolism, and cutin, suberin, and wax biosynthesis pathways [4]. Indeed, allopolyploids have caused a remarkable increase in NLR dosage. *Dalbergioids* present an ideal case where two additional genera *Stylosanthes* and *Aeschynomene,* are polyploidy. We evaluated the expression of *NLR* genes in 10 diploid species of *Aeschynomene,* where relatively similar expression was observed in all species. The availability of reference allotetraploid genomes for this genus will provide further insights into the effect of polyploidy on NLRome evolution.

This study provides evidence of the reduced conservation of *NLR* genes across *Dalbergioids.* Comprehensive phylogeny suggests the presence of seven subgroups of *NLR* genes, including one TNL and six CNL-NLRs. Less variation was observed in a number of helper *NLR* genes (CCR-NLR) in all species of *Dalbergioids*. Duplication patterns were clade-specific, and the majority occurred after speciation resulting in divergent evolution; for example, *N. schottii* revealed a significant increase in the CCG10-*NLR* gene ratio, and the duplication assay also suggests a recent duplication of CCG10-NLR. Overall, our findings provide new insights that will help in the identification and characterization of novel resistance genes in the member species of *Dalbergioids*.

## Figures and Tables

**Figure 1 genes-14-00377-f001:**
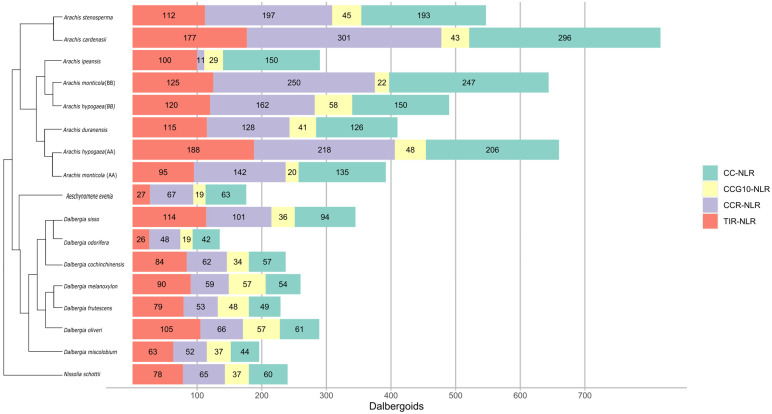
Horizontal bar plots illustrate the proper placement of four classes of NLRs genes indicated with different colors in different species of *Adesmia*, *Dalbergia*, and *Pterocarpus* clades.

**Figure 2 genes-14-00377-f002:**
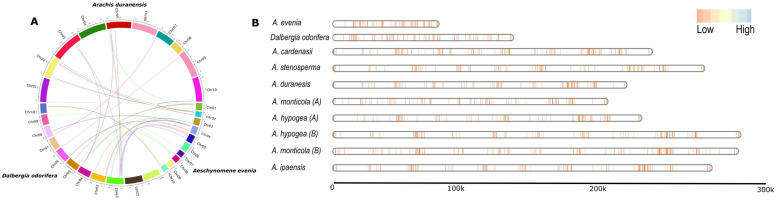
Landscape of *NLR* genes using synteny and gene density analysis. (**A**) Synteny analysis of *NLR* genes from three species *A. duranensis*, *A. evenia,* and *D. odorifera.* (**B**) The NLRs genes are located on chromosomes denoted in vertical blue and red lines, inferring the gene density map between genome assemblies of *Arachis, Dalbergia,* and *A. evenia* species. These plot show placement of genes on linearized chromosomes that are joined together using a bin size of 5 kb.

**Figure 3 genes-14-00377-f003:**
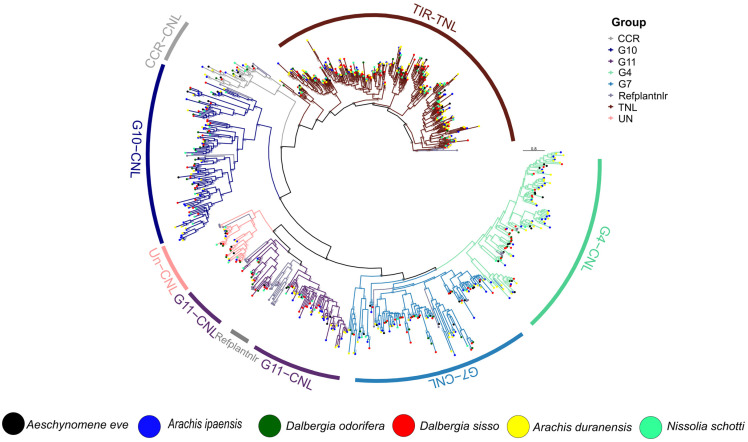
Phylogenetic reconstruction of *NLR* gene identified from *Dalbergioids* species.

**Figure 4 genes-14-00377-f004:**
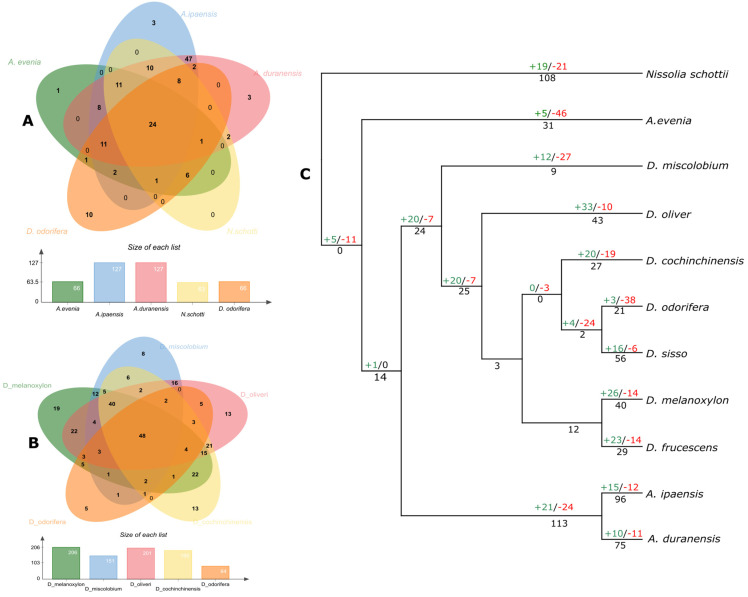
Gene orthologs and number of gene gains and loss analysis: (**A**,**B**) Venn diagrams illustrate the distribution of a number of shared and common genes regarding A- and B-related genome species. (**C**) Each node represents the number of genes, for instance, gene duplication is indicated in black font, and gene gain and loss are shown in green and red color, respectively.

**Figure 5 genes-14-00377-f005:**
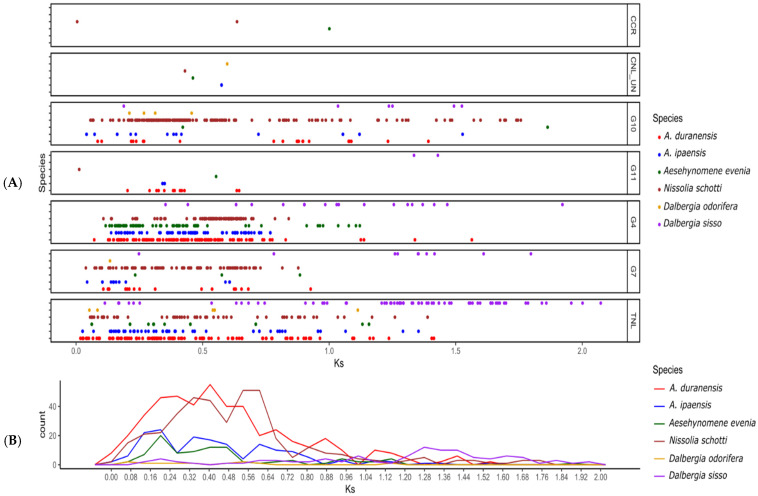
Evaluation of historical *NLR* gene duplication in the genus of *Dalbergia*. All six species belonging to *Dalbergioids* with Ks values of their paralogs are represented. (**A**) X and Y axis indicating the Ks values and their frequencies. (**B**) Generalized *Dalbergioids* duplication pattern. (**B**) These boxplots show the Ks values between *Dalbergia* and *Arachis* species. A middle line represents the median of Ks.

**Figure 6 genes-14-00377-f006:**
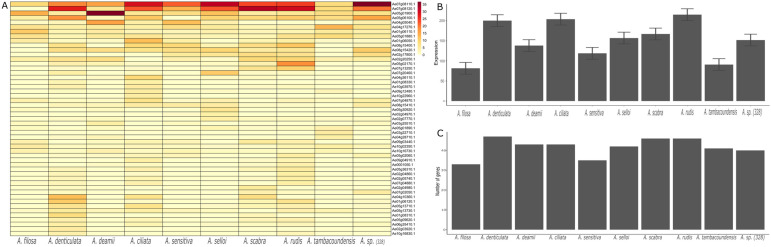
Expression of *NLR* genes in *Aeschynomene* species. (**A**) The heatmap plot represents the expression of 50 *NLR* genes across 10 species of *Aeschynomene* during the rooting stage (day 7). (**B**) The bar plots show the variable cumulative gene expression of NLR in different species under the stress condition of drought. (**C**) The bar plot represents the number of genes expressed during this stage in different species.

## Data Availability

All the datasets generated in this study are attached in Appendix A files.

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
