# Peer review of "Dynamic Evolution of NLR Genes in Dalbergioids"

_genes, 2023, doi:10.3390/genes14020377_

Round 1

Reviewer 1 Report

 Dear the editor,

 The submitted manuscript provides the gene information of NLR family in Dalbergioids and discusses about the evolution; expansion/contraction of the number of NLR genes in the group. Since the NLR genes have fundamental biological functions and the variation of members in the gene family is closely related with diversification of the species, the manuscript has a potential value for the scientific filed.

I have some comments for improving the manuscript.

First, since the number of NLR genes in this study totally depends on the published genome data and transcriptome data, the accuracy in the estimation of NLR gene number relies on the data used in the study. So, the authors should provide the information how accurate the estimation of gene number in this manuscript. Without such information, no readers are able to evaluate the provided data and the arguments in the manuscript.

Second, expansion/contraction of the number of NLR genes among the species in this study would be the interest phenomenon. However, the interpretation would be done more carefully. As the authors discussed in the Discussion section, the global decrease in the gene numbers has been suggested in D. odorifera. The case of D. odorifera raises a question whether the expansion/contraction of the number of genes among the species are specific to NLR or non-specific? Please answer the question.

Last, the data on the gene expression, in particular, Figure 6 seems unrelated to the main theme in the manuscript. In addition, it is not described how the variable cumulative gene expression and the number of expressed genes are determined in the manuscript. The part on gene expression should be rewritten.

Minor points

1.     Line 180, class TIR-NLR exhibits the highest contribution among other classes, on average 46% TIR-NLR. How was the average calculated?

2.     Line 186, Fegure 1 legend is separated from the figure.

3.     Figure 5: (A) and (B) are not indicated.

Author Response

Response to the comment made by Reviewer 1

Dear the editor,

Comment No 1: The submitted manuscript provides the gene information of NLR family in Dalbergioids and discusses about the evolution; expansion/contraction of the number of NLR genes in the group. Since the NLR genes have fundamental biological functions and the variation of members in the gene family is closely related with diversification of the species, the manuscript has a potential value for the scientific filed.

Response 1: My team sincerely pays thanks to the reviewer for raising important points and acknowledge the value of the research questions addressed in this manuscript.

I have some comments for improving the manuscript.

First, since the number of NLR genes in this study totally depends on the published genome data and transcriptome data, the accuracy in the estimation of NLR gene number relies on the data used in the study. So, the authors should provide the information how accurate the estimation of gene number in this manuscript. Without such information, no readers are able to evaluate the provided data and the arguments in the manuscript.

Response 2 : Indeed the accuracy of these results greatly depends upon the quality of genome and transcriptome utilized in this study. For this reason, well accepted reference genomes were studied in this study. Secondly, transcriptome that give more than 90% assembly percentage were kept in this analysis and the datasets with below threshold (90%) were not incorporated in this study. Secondly, the reviewer rightly raised the validity of number of NLR genes identified and characterized. This is the founding study that utilized state of the art NLR mining tool called as NLRtracker. The efficiency and validity provided by NLRtracker are unrivaled to any other contemporary tools. Recently published article of NLRtracker has compared the efficacy of the results with contemporary tools 1. In addition we counter checked the number of genes by doing manual clustering and phylogeny of NLR genes. And to this date we have screened 55 genomes of family Fabaceae but we did not find any discrepancies.

  1. Kourelis, J., Sakai, T., Adachi, H. & Kamoun, S. RefPlantNLR is a comprehensive collection of experimentally validated plant disease resistance proteins from the NLR family. PLoS Biol. (2021) doi:10.1371/journal.pbio.3001124.
  2. Hong, Z. et al. The chromosome-level draft genome of Dalbergia odorifera. Gigascience 9, (2020).

Second, expansion/contraction of the number of NLR genes among the species in this study would be the interest phenomenon. However, the interpretation would be done more carefully. As the authors discussed in the Discussion section, the global decrease in the gene numbers has been suggested in D. odorifera. The case of D. odorifera raises a question whether the expansion/contraction of the number of genes among the species are specific to NLR or non-specific? Please answer the question.

Response 3:

We again thanks the reviewer for their remarkable question raised here. Our analysis strongly suggest that there is a dramatic contraction in the number of NLR genes in D. odorifera. Recently, the chromosomal anchored assembly of D. odorifera confirms that this contraction is not specific to NLR gene family, but this contraction occurred globally. Zhou et al has performed global gene family birth and death analysis on D. odorifera (Figure 2C) where they suggested dramatic contraction of 214 gene families unlike other Fabaceae members 2.

Last, the data on the gene expression, in particular, Figure 6 seems unrelated to the main theme in the manuscript. In addition, it is not described how the variable cumulative gene expression and the number of expressed genes are determined in the manuscript. The part on gene expression should be rewritten.

Response 4 : We thank the reviewer of paying attention to important details of this manuscript, the basic purpose of incorporating expression data was to study the whether the identified genes are active expressed and if yes then what is their extent in different conditions In this way researcher from diverse fields can one day connect their findings with this study; I strongly suggest that this part should remain in MS, For assist readers whole section is rephrased now and missing information is added in the material and methods

Minor points

  1. Line 180, class TIR-NLR exhibits the highest contribution among other classes, on average 46% TIR-NLR. How was the average calculated?

>>Line ??? The average of each subclass was calculated by adding all percentage of all TIR NLR identified from all species divided by number of species

  1. Line 186, Fegure 1 legend is separated from the figure.

>> Line 187 Modified as suggested

  1. Figure 5: (A) and (B) are not indicated.

>> Figure 5; Modified as suggested

Reviewer 2 Report

Dear Authors,

The article "Dynamic evolution of NLR genes in Dalbergioids" is a great contribution for the researchers who are following the NLR role in disease resistance.

Need a few clarifications on the genome data used in the article:

1. Plant species used for genome analysis (NLR) are from the same geographical location?

2. Origin of the NLR, is it from nuclear/mitochondria/plastid genome?

Regards

K. N. Chandrashekara

Author Response

Comments and responses for reviewer 2

Dear Authors,

The article "Dynamic evolution of NLR genes in Dalbergioids" is a great contribution for the researchers who are following the NLR role in disease resistance.$=

  Response 1:

We thank the reviewers for their encouragement. The main purpose was to provide a comprehensive report about the evolution of NLR genes to identifiable species with novel resistance sources. That will help in the development of resistance using conventional breeding approaches.

Need a few clarifications on the genome data used in the article:

  1. Plant species used for genome analysis (NLR) are from the same geographical location?

 Response 2. Plant species for genome analysis are from diverse geographical region they belong to three different clades.

  1. Origin of the NLR, is it from nuclear/mitochondria/plastid genome?

Response 3

The origin of these NLR genes is nuclear. They have not been reported from chloroplast

Round 2

Reviewer 1 Report

Dear the editor,

 The revised manuscript includes additional information relevant to my previous comments.  I have no more suggestions.

Best